**Data Availability Statement:** Data cannot be shared publicly due to sensitive patient information. Data are available from the Taipei Veterans General Hospital Institutional Data Access

# Factor analysis for the clustering of cardiometabolic risk factors and sedentary behavior, a cross-sectional study

**Tsung-Ying Tsai**[1], **Pai-Feng Hsu**[1,2,3,4], **Chung-Chi Lin**[1,4], **Yuan-Jen Wang**[1,4], **Yaw-Zon Ding**[1,4], **Teh-Ling Liou**[1,4], **Ying-Wen Wang**[1,4], **Shao-Sung Huang**[1,2,4], **Wan-Leong Chan**[1,2,4], **Shing-Jong Lin**[1,2,4,5,6], **Jaw-Wen Chen**[1,2,4], **Hsin-Bang Leu**[1,2,4,5,6] *

1 Division of Cardiology, Department of Medicine, Taipei Veterans General Hospital, Taipei, Taiwan, 2 Healthcare and Management Center, Taipei Veterans General Hospital, Taipei, Taiwan, 3 Department of Public Health, National Yang-Ming University, Taipei, Taiwan, 4 School of Medicine, National Yang-Ming University, Taipei, Taiwan, 5 Cardiovascular Research Center, National Yang-Ming University, Taipei, Taiwan, 6 Institute of Clinical Medicine, National Yang-Ming University, Taipei, Taiwan

* hsinbangleu@gmail.com

## Abstract

### Background

Few studies have reported on the clustering pattern of CVD risk factors, including sedentary behavior, systemic inflammation, and cadiometabolic components in the general population.

### Objective

We aimed to explore the clustering pattern of CVD risk factors using exploratory factor analysis to investigate the underlying relationships between various CVD risk factors.

### Methods

A total of 5606 subjects (3157 male, 51.5±11.7 y/o) were enrolled, and 14 cardiovascular risk factors were analyzed in an exploratory group (n = 3926) and a validation group (n = 1676), including sedentary behaviors.

### Results

Five factor clusters were identified to explain 69.4% of the total variance, including adiposity (BMI, TG, HDL, UA, and HsCRP; 21.3%), lipids (total cholesterol and LDL-cholesterol; 14.0%), blood pressure (SBP and DBP; 13.3%), glucose (HbA1C, fasting glucose; 12.9%), and sedentary behavior (MET and sitting time; 8.0%). The inflammation biomarker HsCRP was clustered with only adiposity factors and not with other cardiometabolic risk factors, and the clustering pattern was verified in the validation group.

/ Ethics Committee (irbopinion@vghtpe.gov.tw) or via the corresponding author (hsinbangleu@gmail.com) for researchers who meet the criteria for access to confidential data.

**Funding:** This study is funded by the Research foundation of cardiovascular medicine. The primary recipient of research fund is Dr. Hsin-Bang Leu. The funders had no role in study design, data collection and analysis, decision to publish, or preparation of the manuscript.

**Competing interests:** The authors have declared that no competing interests exist.

## Conclusion

This study confirmed the clustering structure of cardiometabolic risk factors in the general population, including sedentary behavior. HsCRP was clustered with adiposity factors, while physical inactivity and sedentary behavior were clustered with each other.

## Background

Despite significant advances over the past decades, atherosclerotic cardiovascular diseases (CVD) remain the most significant cause of mortality worldwide [1]. CVD is a complex disease, and decades of research has recognized that CVD risk factors are clustered in specific patterns imply common underlying disease processes [2]. Metabolic abnormalities such as central obesity, insulin resistance, dyslipidemia, and high blood pressure are highly involved in the pathogenesis of CVD and are recognized as metabolic syndrome. Metabolic syndrome has variable presentation and is considered as a high-risk factor for CVD [3]. However, with the evolving understanding of the pathophysiology of atherosclerosis, risk factors are being added to the ever-expanding battery of CV risk factors.

Sedentary behavior (SB) and physical activity (PA) has been established as an independent risk factor for CVD on its own [4–7]. The current professional guidelines recommend avoiding SB and maintaining adequate PA, which are considered major goals in public health policy [8]. SB has also been linked to reduced triglyceride metabolism, insufficient antioxidant production, and glucose intolerance in several animal studies [9]. The underlying pathophysiologic link between SB and other CVD risk factors are deeply intertwined. There has been a great number of studies demonstrating the association between SB and tradition CVD risk factors. For example, in the landmark NHANES 2003–2006 study. total sedentary time was detrimentally associated with several biomarkers including waist circumference, HDL-cholesterol, C-reactive protein, triglycerides, insulin, and insulin resistance. However, while this study demonstrated the close relationship between SB and inflammation, the information of other important risk factors including total cholesterol, UA, HbA1C, were missing [10]. In a later meta-analysis, Edwardson et al. demonstrated that patients with longer sedentary time have greater odds of having metabolic syndrome [11]. However, it is impossible to delineate the association between SB and a particular component of metabolic syndrome. Hence, although the importance of SB is undeniable, there have only been a few studies have investigated the clustering relationship between SB and other risk factors, and there is insufficient information to explain the variances of metabolic abnormalities observed [10–14]. Furthermore, the definition of SB has not been well established in previous studies [15].

Exploratory factor analysis is a statistical method of data reduction that allows investigators to overcome the analytical difficulty posed by the vast number of risk variables by demonstrating the underlying relationships between different risk factors [16]. Factor analysis has been performed on many different populations but PA and SB have not been included in most major studies [17–19]. In addition, CVD has long been considered a systemic inflammatory disease. Inflammatory biomarkers such as high-sensitive C-reactive protein (HsCRP) are used to reflect disease severity and guide treatment strategies [20, 21]. Therefore, the current study investigates the clustering relationship of SB, cardiometabolic components, and inflammatory biomarkers in the general population of Taiwan.

## Methods

### Study population

We derived our data from the VGH-HEALTHCARE study, which is a prospective cohort study to evaluate the impact of PA and SB on long-term outcomes [22]. Adult subjects who received a comprehensive health examination at the Healthcare Center of Taipei Veterans General Hospital from February 2015 to July 2019 were invited to join this study. In brief, the healthcare center of Taipei Veterans general hospital provides elective, self-paid health examination services to all individuals who wish to receive health examination to identify undetected conditions for primary prevention purposes. We included patients without significant symptoms or illness and excluded those who refused to participate, whose exam revealed an acute illness, or had a chronic condition that require regular follow up such as active cancer, heart failure, coronary artery disease or stroke. We believe that data from our study population could provide vital information on primary prevention from the general population perspective. Demographic data, biochemical blood tests, and information on PA and SB were collected.

VGH-HEALTHCARE is an ongoing prospective study, so information on long-term outcomes is currently not available. The present study serves as a cross-section analysis to investigate the relationship between baseline cardiometabolic factors, SB, and inflammation biomarkers. The enrolled subjects were divided into two groups with similar baseline characteristics, with 70% in the exploratory group and 30% in the validation group. All participants provided a written consent. This study was conducted in concordance with the Declaration of Helsinki and was approved by the Internal Research Board of Taipei Veterans General Hospital. All information was obtained after receiving informed consent from the study participants.

### Clinical assessments and biochemical parameters

The baseline information included age, sex, body height, body weight, body mass index (BMI), waist circumference, monthly income, education level, alcohol drinking behavior, and smoking status. The collected medical history included hypertension, type 2 diabetes, and hyperlipidemia. After an overnight fast, a TBA-c16000 automatic analyzer (Toshiba Medical Systems, Tochigi, Japan) was used to measure biochemical parameters, including fasting glucose, hemoglobin A1C (HbA1C), total cholesterol (TC), triglycerides (TG), low-density lipoprotein cholesterol (LDL-C), high-density lipoprotein cholesterol (HDL-C), aspartate aminotransferase (AST), alanine aminotransferase (ALT), total bilirubin, uric acid (UA), and high-sensitivity C-reactive protein (hsCRP) [23, 24].

### Physical activity and sedentary behavior

PA levels and sedentary status were assessed using the validated Chinese version of the International Physical Activity Questionnaire–Short Form (IPAQ-SF) [25, 26]. All of the included patients filled out the questionnaires within one hour at noon with the assistance of specially trained nurses. The IPAQ-SF includes the number of days and the duration of vigorous, moderate, and walking activities during the previous week [27, 28]. The IPAQ-SF enables the calculation of metabolic equivalents (MET minutes per week), which are derived by assigning standardized MET values of 3.3, 4, and 8 for walking, moderate-intensity activity, and vigorous-intensity activity, respectively. These data were quantified, and an estimated metabolic equivalent of a task for each individual was classified as high, moderate, or low PA according to the IPAQ-SF score. The total daily sitting time was also collected for all participants.

## Statistical analysis

The data are expressed as the mean ± standard deviation for normally distributed continuous variables and as number(percentage) for categorical variables. Demographic characteristics and biochemical variables were compared using student's t-tests and the Mann–Whitney U test for the comparison of continuous variables, while chi-squared tests were used for categorical variables. Statistical significance was considered as $P < 0.05$. All statistical analyses were carried out using SPSS 20.0 software (IBM, Inc. Chicago, USA).

We performed an exploratory factor analysis to determine the clustering of cardiovascular risk factors, PA, and SB. The detailed statistical method has been described previously [29]. In brief, exploratory factor analysis is a statistical method designed to reveal the inter-correlations between the analyzed variables by reducing the collected variables into summary factors while retaining as much of the variance in the original variables as possible. There are three main steps of factor analysis: (1) extraction of factors (principal component analysis (PCA)); (2) rotation of factors to obtain a simple structure that can be easily interpreted; and (3) naming and interpreting each factor based on estimated values for factor loadings.

We used PCA to identify the principal components that reflect a group of variables that act together on a common pathophysiological process. We used an eigenvalue >1 as the extraction threshold, which was calculated as the sum of the squared factor loadings, which is a measurement of the amount of variation in the total sample accounted for by each factor. We then used orthogonal rotation (varimax rotation) to obtain factor loadings. We used an absolute loading value of >0.4 to interpret the factor pattern, which has been used by previous major factor analysis studies [30, 31].

In the primary analysis, factors were derived from 14 potential CVD risk factors: BMI, waist circumference, systolic blood pressure (SBP), diastolic blood pressure (DBP), fasting blood glucose, HbA1C, TC, HDL-C, LDL-C, TG, UA, HsCRP, MET, and daily sitting time. We first tested pair-wise correlations. We defined a KMO value greater than 0.6 and a significant Bartlett's test of sphericity (P<0.001) as an indication for sampling adequacy and a lack of an identity matrix. Because it is difficult to find another population with the same low risk who had also completed the questionnaires as well as a biochemical study including CRP, we consider that it was reasonable to divide the whole population into two groups, one for training and one for validation to avoid overfitting the developed model. We randomly assigned 70% of the total population to the training group while the remaining 30% were assigned to the validation group. In statistics for data mining, it is a common method that training data is often a subset of the total data set, and the test set is a subset of the test-trained model. Similar analyses have been reported previously. For example, in Hravnak's machine learning algorithms data were divided into Block 1 for the ML training/cross-validation set and Block 2 for the test set [32]. Furthermore, Goodman et al. used factor analysis for the cardiovascular clustering risk, and they selected 20% of the cases as an exploratory sample and the remaining 80% cases were used as a validation sample [30]. Because our current study aimed to investigate the cluster risk including inflammation makers and SB information, it is reasonable to test and validate sample using the same number of variables.

## Results

A total of 5606 subjects (3157 male, age 51.5±11.7) were enrolled in this study. The enrollees were divided into a training group (N = 3926) and a validation group (N = 1676). The participants' baseline characteristics, biochemical data, PA, and SB are shown in Table 1. The enrolled population consists of healthy middle age Taiwanese people. There was no significant difference between the demographic and biochemical data of the two groups.

**Table 1. Baseline characteristics, comorbidities, and physical activity of the two groups.**

| | Training group (n = 3926) | Validation group (n = 1676) | p value |
|---|---|---|---|
| Male, n (%) | 947.0 (56.7) | 2,201.0 (56.5) | 0.860 |
| Age, years | 51.58 ± 11.65 | 51.54 ± 11.81 | 0.902 |
| Height, cm | 165.47 ± 8.33 | 165.64 ± 8.41 | 0.491 |
| Weight, kg | 66.25 ± 12.88 | 66.23 ± 12.94 | 0.971 |
| BMI, kg/m2 | 24.07 ± 3.58 | 24.02 ± 3.64 | 0.626 |
| Waist circumference, cm | 84.47 ± 10.04 | 84.46 ± 10.11 | 0.959 |
| Smoking, n (%) | 343.0 (21.1) | 737.0 (19.4) | 0.130 |
| Drinking, n (%) | 552.0 (34.1) | 1,271.0 (33.4) | 0.652 |
| Dyslipidemia, n (%) | 205.0 (12.7) | 514.0 (13.6) | 0.411 |
| Diabetes, n (%) | 122.0 (7.6) | 250.0 (6.6) | 0.193 |
| Hypertension, n (%) | 284.0 (17.6) | 671.0 (17.7) | 0.953 |
| SBP, mmHg | 120.39 ± 16.89 | 120.22 ± 17.06 | 0.736 |
| DBP, mmHg | 76.80 ± 10.66 | 76.65 ± 10.73 | 0.657 |
| Cholesterol, mg/dL | 202.34 ± 37.61 | 203.48 ± 38.10 | 0.315 |
| Triglyceride, mg/dL | 120.08 ± 73.10 | 119.50 ± 72.79 | 0.792 |
| Uric acid, mg/dL | 6.24 ± 1.52 | 6.28 ± 1.55 | 0.487 |
| HDL, mg/dL | 50.68 ± 13.89 | 50.24 ± 13.54 | 0.286 |
| LDL, mg/dL | 127.27 ± 34.15 | 128.56 ± 34.41 | 0.209 |
| HbA1c, % | 5.65 ± 0.74 | 5.65 ± 0.75 | 0.997 |
| GLU, mg/dL | 93.54 ± 21.84 | 93.31 ± 21.80 | 0.719 |
| AST, U/L | 24.22 ± 10.97 | 24.61 ± 13.73 | 0.311 |
| ALT, U/L | 27.20 ± 19.60 | 27.89 ± 25.60 | 0.336 |
| Total Bilirubin, mg/dL | 1.11 ± 0.55 | 1.12 ± 0.50 | 0.931 |
| Creatinine, mg/dL | 0.87 ± 0.37 | 0.87 ± 0.23 | 0.936 |
| hsCRP, mg/dL | 0.19 ± 0.36 | 0.19 ± 0.34 | 0.924 |
| sitting time, min per day | 388.63 ± 186.27 | 393.26 ± 190.37 | 0.402 |
| METs, per week | 1,524.33 ± 1,837.33 | 1,554.67 ± 1,902.77 | 0.581 |

ALT = alanine aminotransferase; AST = aspartate aminotransferase; BMI = body mass index; GLU = serum glucose; hsCRP = high sensitive C-reactive protein; HDL = high density lipoprotein; LDL = low density lipoprotein; SBP = systolic blood pressure; DBP = diastolic blood pressure; MET = metabolic equivalent; HbA1C = hemoglobin A1C.

The correlation among the 14 variables among all subjects is shown in Table 2. Sitting time demonstrated a negative correlation with MET (Pearson correlation coefficient (r) = -0.165, $P < 0.001$). MET value was significantly negatively correlated with waist circumference, blood pressure, cholesterol, triglyceride, LDL and sitting time, but positively correlate with HDL. We have further categories subjects into low, moderate, and high PA group and we found that subjects with low activity have unfavorable lipid profiles and higher baseline inflammatory makers, supporting the connection between higher cardiovascular risk to low PA (Table 3). However, although sitting time is negatively correlated with MET values, the correlation between lipid profiles only existed in HDL. It is not surprising because sitting time is only one factor among the definition of SB and total MET activity estimation was generated from more activity's information. To explore the effect of gender and age of factor clustering pattern, we re-analyzed the correlation, taking into consideration gender and age which showed similar factor clustering in all subgroups (S1–S4 Tables).

Tables 3 and 4 show the factor analysis results of the training group and the validation group. In the training group (Table 3), PCA identified five factors with an eigenvalue >1. The

**Table 2. Correlation of cardiovascular risk factors.**

| | Age | BMI | WaistC | SBP | DBP | Cholesterol | TG | UA | HDL | LDL | A1c | GLU | sitting time | METs |
|---|---|---|---|---|---|---|---|---|---|---|---|---|---|---|
| Age | 1 | .107† | .170† | .271† | .122† | .087† | .051† | .034* | -.035* | .060† | .289† | .231† | -.180† | .063† |
| BMI | -- | .1 | .861† | .340† | .321† | -.001 | .350† | .387† | -.444† | .091† | .260† | .285† | .022 | -.003 |
| Waist Circum | -- | -- | 1 | .334† | .316† | -.008 | .353† | .414† | -.463† | .082† | .272† | .302† | .001 | -.028* |
| SBP | -- | -- | -- | 1 | .729† | .040* | .177† | .216† | -.159† | .056† | .180† | .201† | -.065† | .073† |
| DBP | -- | -- | -- | -- | 1 | .061† | .216† | .246† | -.186† | .088† | .126† | .158† | -.014 | .033* |
| Cholesterol | -- | -- | -- | -- | -- | 1 | .191† | .103† | .226† | .894† | .023 | -.022 | -.035* | -.031* |
| Triglyceride | -- | -- | -- | -- | -- | -- | 1 | .297† | -.447† | .052† | .252† | .311† | .014 | -.059† |
| UricAcid | -- | -- | -- | -- | -- | -- | -- | 1 | -.356† | .167† | .081† | .064† | .018 | .030* |
| HDL | -- | -- | -- | -- | -- | -- | -- | -- | 1 | -.046† | -.192† | -.226† | -.046† | .032* |
| LDL | -- | -- | -- | -- | -- | -- | -- | -- | -- | 1 | .020 | -.037* | -.005 | -.039* |
| HbA1c | -- | -- | -- | -- | -- | -- | -- | -- | -- | -- | 1 | .778† | -.049† | -.007 |
| GLU | -- | -- | -- | -- | -- | -- | -- | -- | -- | -- | -- | 1 | -.024 | .001 |
| sitting time | -- | -- | -- | -- | -- | -- | -- | -- | -- | -- | -- | -- | 1 | -.165† |
| METs | -- | -- | -- | -- | -- | -- | -- | -- | -- | -- | -- | -- | -- | 1 |

*0.05≧P≧0.01,

† 0.01≧P;

BMI = body mass index; GLU = serum glucose; HDL = high density lipoprotein; LDL = low density lipoprotein; SBP = systolic blood pressure; DBP = diastolic blood pressure; MET = metabolic equivalent; HbA1C = hemoglobin A1C.

combined factors explained 69.43% of the variance among the original 14 factors. BMI, WC, HsCRP, TG, UA, and HDL-C were grouped together in the first common factor, the adiposity factor, which is similar to the metabolic syndrome criteria put forth by the NCEP/ATP III guidelines [33]. The inflammation biomarker hsCRP, which represents the underlying inflammation status of a subject, was shown within the adiposity group. This factor explained 21.30% of the total variance. The second common factor was the lipid factor, which contained LDL-C and total cholesterol and explained approximately 13.97% of the total variance.

The third common factor, the blood pressure factor, consisted of SBP and DBP and accounted for 13.03% of the variance. The fourth common factor, the glucose factor, included fasting blood glucose and HbA1C and accounted for 12.91% of the variance. The final common factor, the SB factor, consisted of both the MET and the daily sitting time, which explained 7.96% of the variance.

For the results of the validation samples (Table 5), the PCA also identified five factors. The combined factors explained 69.48% of the variance in the original 14 factors. The adiposity factor, lipid factor, blood pressure factor, glucose factor, and activity factor explained 20.26%, 13.96%, 13.81%, 13.03, and 8.43% of the variance, respectively. Figs 1 and 2 show the component plots with factor diagrams from the PCA with varimax rotation. Both groups demonstrated a consistent clustering of risk factors.

To explore the composition of factor clustering in different gender and age, we performed age and sex-stratified factor analysis. The results showed a similar clustering pattern of risk factors in both gender and age groups (S1–S4 Tables).

## Discussion

This single-center cross-sectional analysis examined 5606 healthy Asian adults and demonstrated that complex clustering cardiometabolic factors can be divided into five factor clusters:

**Table 3. Baseline characteristics, comorbidities, and physical activity of different physical activity groups.**

| | "Low (n = 2398)" | "Moderate (n = 2154)" | "High (n = 844)" | p value |
|---|---|---|---|---|
| Age(y/o) | 50.18 ± 11.42 | 52.60 ± 11.89 | 52.75 ± 12.01 | < .001 |
| Height | 165.29 ± 8.29 | 165.67 ± 8.56 | 166.28 ± 8.19 | 0.010 |
| Weight | 66.23 ± 13.38 | 66.08 ± 12.73 | 66.63 ± 12.01 | 0.568 |
| BMI | 24.12 ± 3.81 | 23.95 ± 3.51 | 24.00 ± 3.34 | 0.283 |
| Waist circumference | 84.83 ± 10.49 | 84.32 ± 9.90 | 83.81 ± 9.34 | 0.037 |
| Smoking (n, %) | 533 (22.0) | 386 (17.8) | 159 (18.9) | 0.001 |
| Drinking (n, %) | 847 (35.1) | 682 (31.6) | 290 (34.5) | 0.036 |
| Dyslipidemia (n, %) | 307 (12.8) | 303 (14.1) | 109 (12.9) | 0.424 |
| Diabetes (n, %) | 160 (6.7) | 152 (7.1) | 59 (7.0) | 0.868 |
| Hypertension (n, %) | 396 (16.5) | 415 (19.3) | 143 (16.9) | 0.043 |
| SBP | 119.15 ± 16.55 | 120.72 ± 17.36 | 122.32 ± 17.11 | < .001 |
| DBP | 76.48 ± 10.69 | 76.71 ± 10.78 | 77.32 ± 10.56 | 0.143 |
| Cholesterol | 204.33 ± 38.48 | 202.47 ± 37.93 | 201.58 ± 36.37 | 0.111 |
| Triglyceride | 124.99 ± 75.70 | 117.90 ± 73.99 | 109.00 ± 59.39 | < .001 |
| UricAcid | 6.27 ± 1.57 | 6.26 ± 1.50 | 6.29 ± 1.54 | 0.883 |
| HDL | 49.87 ± 13.40 | 50.48 ± 13.86 | 51.57 ± 13.69 | 0.008 |
| LDL | 129.45 ± 34.75 | 127.43 ± 34.40 | 126.51 ± 32.82 | 0.046 |
| A1c | 5.64 ± 0.76 | 5.67 ± 0.74 | 5.63 ± 0.74 | 0.360 |
| GLU | 93.36 ± 22.00 | 93.49 ± 21.69 | 93.17 ± 21.44 | 0.935 |
| AST | 24.35 ± 13.48 | 24.41 ± 12.99 | 25.07 ± 11.15 | 0.361 |
| ALT | 28.25 ± 22.86 | 27.31 ± 26.94 | 26.89 ± 17.47 | 0.251 |
| Total Bilirubin | 1.12 ± 0.54 | 1.12 ± 0.49 | 1.11 ± 0.49 | 0.900 |
| Creatinine | 0.86 ± 0.32 | 0.88 ± 0.27 | 0.88 ± 0.18 | 0.014 |
| hsCRP | 0.22 ± 0.39 | 0.18 ± 0.32 | 0.15 ± 0.29 | 0.004 |
| sitting time (min per day) | 421.64 ± 197.75 | 379.51 ± 178.33 | 339.07 ± 163.14 | < .001 |
| METs (per week) | 374.97 ± 488.09 | 1,494.85 ± 663.30 | 4,901.42 ± 2,318.10 | < .001 |

ALT = alanine aminotransferase; AST = aspartate aminotransferase; BMI = body mass index; GLU = serum glucose; hsCRP = high sensitive C-reactive protein; HDL = high density lipoprotein; LDL = low density lipoprotein; SBP = systolic blood pressure; DBP = diastolic blood pressure; MET = metabolic equivalent; HbA1C = hemoglobin A1C.

the adiposity factor (waist circumference, BMI, TG, HDL, and UA), the blood pressure factor (SBP and DBP), the lipid factor (TC and LDL), the glucose factor (fasting glucose and HbA1C), and the PA factor. These factors explained 21.97%, 13.97%, 13.30%, 12.91%, and 7.96% of the total variance, respectively. Systemic inflammation was linked to the adiposity factor, while SB and PA were clustered together and formed an independent CVD risk factor. Multiple cardiometabolic factors were involved in the development and progression of athero-sclerotic cardiovascular disease.

This study is the first to investigate the relationship of cardiometabolic factors, systemic inflammation, and sedentary information simultaneously in the general population of Taiwan. We also demonstrated that the PA factors do not cluster with traditional CV risk factors. Our results suggest that physical inactivity may exert its effect on cardiovascular disease in an independent and unique way. This result may prompt future researchers to explore the possible pathophysiologic mechanism behind the independent effect of the PA level. Our findings could also provide important evidence that adiposity is linked to baseline inflammation, which explained nearly 20% of the variance in subjects without CVD.

**Table 4. Factor analysis of the training group.**

| | Component | | | | |
|---|---|---|---|---|---|
| | 1 | 2 | 3 | 4 | 5 |
| Waist Circumference | .794 | | | | |
| BMI | .788 | | | | |
| HDL | -.719 | | | | |
| Uric acid | .686 | | | | |
| Triglyceride | .610 | | | | |
| HsCRP | .492 | | | | |
| Cholesterol | | .986 | | | |
| LDL | | .942 | | | |
| SBP | | | .918 | | |
| DBP | | | .902 | | |
| GLU | | | | .910 | |
| A1c | | | | .901 | |
| METs (per week) | | | | | .757 |
| sitting time (per day) | | | | | -.713 |
| Eigenvalues | 3.733 | 1.958 | 1.523 | 1.424 | 1.083 |
| Rotation Sums of Squared Loadings (% of Variance) | 21.297 | 13.965 | 13.303 | 12.910 | 7.957 |
| Rotation Sums of Squared Loadings (Cumulative %) | 21.297 | 35.263 | 48.566 | 61.476 | 69.433 |

## Systemic inflammation and cardiovascular risk factors

Increased baseline inflammation is believed to play a crucial role in all stages of the arthero-thrombotic disease process, and treatment strategies to reduce inflammation have been

**Table 5. Factor analysis of the validation group.**

| | Component | | | | |
|---|---|---|---|---|---|
| | 1 | 2 | 3 | 4 | 5 |
| Waist Circumference | .808 | | | | |
| BMI | .797 | | | | |
| HDL | -.740 | | | | |
| Uric acid | .660 | | | | |
| Triglyceride | .504 | | | | |
| HsCRP | .431 | | | | |
| Cholesterol | | .985 | | | |
| LDL | | .946 | | | |
| GLU | | | .914 | | |
| A1c | | | .908 | | |
| SBP | | | | .909 | |
| DBP | | | | .890 | |
| METs (per week) | | | | | .803 |
| sitting time (per day) | | | | | -.677 |
| Eigenvalues | 3.762 | 1.962 | 1.483 | 1.403 | 1.118 |
| Rotation Sums of Squared Loadings (% of Variance) | 20.262 | 13.955 | 13.814 | 13.026 | 8.426 |
| Rotation Sums of Squared Loadings (Cumulative %) | 20.262 | 34.217 | 48.030 | 61.057 | 69.483 |

BMI = body mass index; GLU = serum glucose; HDL = high density lipoprotein; LDL = low density lipoprotein; SBP = systolic blood pressure; DBP = diastolic blood pressure; MET = metabolic equivalent; HbA1C = hemoglobin A1C.

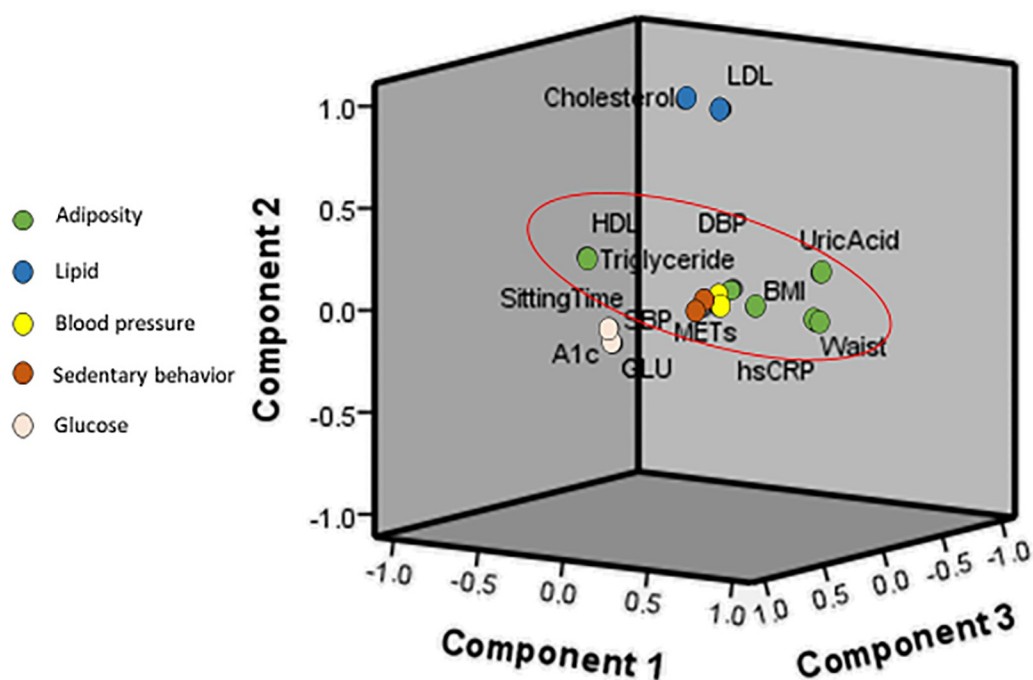

**Fig 1. Component plots of the training group with factor diagrams from principle component analysis with varimax rotation.** BMI = body mass index; GLU = serum glucose; HDL = high density lipoprotein; LDL = low density lipoprotein; SBP = systolic blood pressure; DBP = diastolic blood pressure; MET = metabolic equivalent; HbA1C = hemoglobin A1.

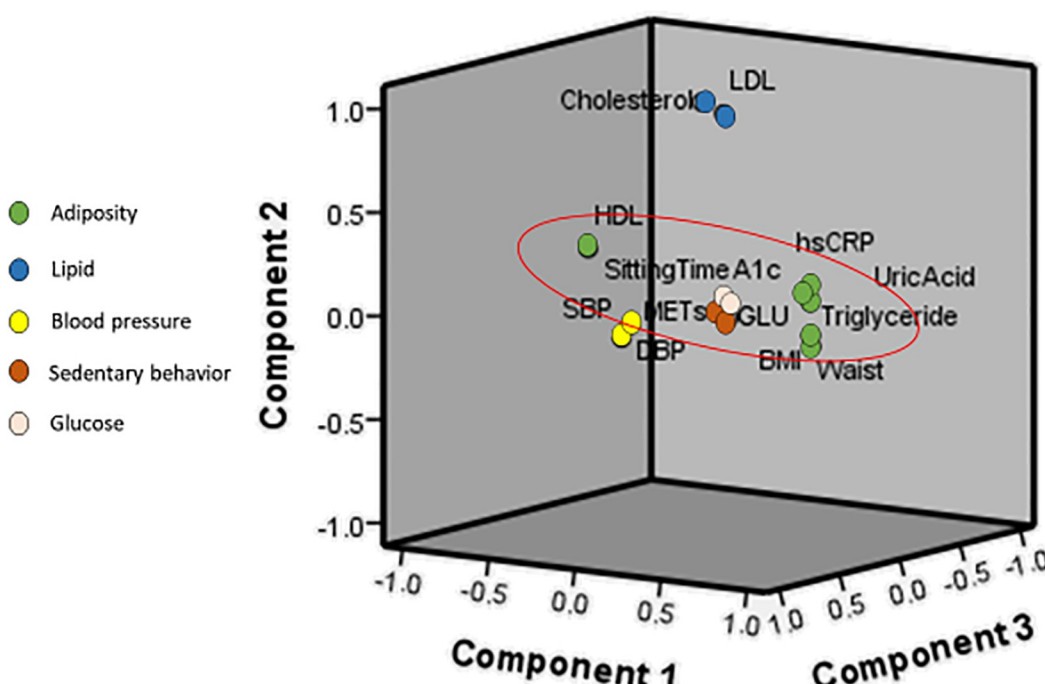

**Fig 2. Component plots of the validation group with factor diagrams from principle component analysis with varimax rotation.** BMI = body mass index; GLU = serum glucose; HDL = high density lipoprotein; LDL = low density lipoprotein; SBP = systolic blood pressure; DBP = diastolic blood pressure; MET = metabolic equivalent; HbA1C = hemoglobin A1.

demonstrated to lower the residual risk of CVD in addition to lipid-lowering medications [20, 21, 34]. We demonstrated that HsCRP was a part of the adiposity factor but was independent of the lipid factor (TC and LDL). The correlation between baseline CRP and LDL has been a controversial topic in previous observations [35]. The JUPITER trial demonstrated that risk reduction with statin therapy is related to the level of CRP, but no such relationship was observed for LDL-C. This suggests that statins reduced hsCRP independently of LDL and that LDL was not related to hsCRP [36, 37].

The current study extended the understanding that inflammation is linked to the adiposity factor and contributes to cardiometabolic clustering variance in the general population of Taiwan. Nicklas et al. showed that the change in hsCRP after weight loss is significantly correlated with changes in total body fat, abdominal adiposity, visceral adiposity, and lipid profiles, especially TG and HDL, but not LDL. This supports the close relationship between baseline inflammation and adiposity [38]. It is also worth mentioning that uric acid was also linked with adiposity in our study. This result is consistent with a previous factor analysis study of 2945 adults from the FIBER study [39]. Uric acid has been recognized as a maker for inflammation and oxidative stress, which underlie the disease processes of gout and CVD [40]. The inclusion of uric acid further supports our finding that insulin resistance and inflammation are features of adiposity.

## Clustering of sedentary behavior and physical inactivity

SB and PA can explain some of the variance in our study, but the proportion of explained variance was smaller than that of other factors. We also demonstrated that daily sitting time has a negative correlation with weekly PA, suggesting a close relationship between them. The correlation between sitting time and PA has not been consistent in previous studies, with some demonstrating that PA and SB are interdependent, while others show an independent relationship [7, 41–43]. This discrepancy can be explained by the fact that a person can be both sedentary and physically active (the Active Couch Potato phenomenon, describes someone who meets the recommendations for physical activity but still sits around for long periods of the day) [44]. Another explanation is the inconsistent definition of SB across previous studies. Many studies used the lowest MET as a definition of SB, which does not require the subject to be actually sitting [15]. In this study, we used the IPAQ questionnaire, which is a widely validated questionnaire that is recommended by the WHO, to evaluate the association between sedentary behavior and weekly physical activity [45, 46]. Our result is compatible with a recent large study with robust methodology by Stamatakis et al., who showed that the daily sitting time and PA of 149,077 middle age adults were well correlated [42].

However, MET and daily sitting time were not significantly correlated with other cardiometabolic risk factors in our study. To evaluate whether PA and SB are truly independent of other risk factors, we divided our subjects into low, mediate, and high-intensity PA groups according to the definition of the IPAQ questionnaire. Table 3 shows the baseline characteristics of all subjects according to PA. Subjects with low PA have longer sitting times, larger waist circumferences, unfavorable lipid profiles (higher TG, higher LDL-C, and lower HDL-C) and higher baseline CRP. This indicates that low PA was still associated with risk factors of cardiovascular disease. Indeed, as shown in Table 3, patients with low PA have multiple factors at the same time, making it difficult to evaluate the importance and correlation of individual risk factors. Factor analysis of this complex data has provided more information to dissect the complex clustering of risk factors.

Table 3 have also showed that nearly 43% of our participants belonged to the low PA categories. Although there might be a selection bias that our population may have higher

socioeconomic status, but lower daily activity, the high prevalence of low PA is not unique to our population. Both global and Taiwanese survey have showed equally alarming rate of insufficiently physically activity [47, 48]. Moreover, the WHO report have shown that over the past 15 years, the levels of insufficient activity did not improve [49].

## Study limitations

There were some limitations to the present study. First, the information on sitting time was self-reported and is thus susceptible to reporting error. However, IPAQ was developed to measure health-related PA in the general population, has been tested extensively, and is now used in many international studies [45]. The sitting time was defined as the time subjects spend sitting while at work, at home, and during leisure time. This may include time spent sitting at a desk, socializing with friends, reading, sitting, or lying down watching television. The screen time spent on individual activities such as TV watching or computer/mobile usage was not recorded individually. Second, all of the participants were recruited from those who received a comprehensive annual or biennial examination at a healthcare center, and all subjects were asymptomatic and had few risk factors. A detailed history of medication was not available, and the possibility of selection bias cannot be excluded. Larger studies are also still needed to investigate whether our findings could be applied to other populations. Third, the study was cross-sectional in nature and provided no outcome data. Because the VGH-HEALTHCARE study is still ongoing, the long-term outcome data will be available when the VGH- HEALTHCARE study completes. Fourth, our study did not include emerging risk factors for CVD such as hematological factors and liver function. However Although these factors are considered to be new cardiovascular markers in some studies, there is little evidence supporting their role in the general population compared with the factors analyzed in the current study. Fifth, we did not apply machine learning method in the analysis of our data. We plan to use machine learning method for the analysis of outcome data after the VGH-HEALTHCARE study is completed. However, we did not consider using machine learning in our current study, for two reasons. First, using machine learning for factor analysis has not been widely accepted for lack of long-term outcome information. Second. To make our results more convincing to the general audience, we chose to investigate the percentage of variance of clustering cardiometabolic risk factors and the association between sedentary behavior and other associated risk factors with a well validated statistical method.

## Conclusion

The complex relationship of cardiometabolic factors, inflammation, and sedentary information among the general population of Taiwan can be divided into five factor clusters: the adiposity factor (waist circumference, BMI, TG, HDL, and UA), the blood pressure factor (SBP and DBP), the lipid factor (TC and LDL), the glucose factor (fasting glucose and HbA1C), and the PA factor, which explained 21.97%, 13.97%, 13.30%, 12.91%, and 7.96% of the total variance, respectively. Our results suggest that systemic inflammation shares the same underlying disease process with metabolic syndrome, while the independent role of PA warrants exploration with future studies.

## Supporting information

**S1 Table. Factor analysis in patients <65 years of age.**
(DOCX)

**S2 Table. Factor analysis in patients >65 years of age.**
(DOCX)

**S3 Table. Factor analysis in male patients.**
(DOCX)

**S4 Table. Factor analysis in female patients.**
(DOCX)

## Author Contributions

**Conceptualization:** Tsung-Ying Tsai, Pai-Feng Hsu, Hsin-Bang Leu.

**Data curation:** Pai-Feng Hsu, Chung-Chi Lin.

**Formal analysis:** Tsung-Ying Tsai, Pai-Feng Hsu, Chung-Chi Lin.

**Investigation:** Tsung-Ying Tsai, Hsin-Bang Leu.

**Methodology:** Tsung-Ying Tsai, Pai-Feng Hsu, Chung-Chi Lin, Yuan-Jen Wang, Shao-Sung Huang, Hsin-Bang Leu.

**Supervision:** Chung-Chi Lin, Yaw-Zon Ding, Teh-Ling Liou, Ying-Wen Wang, Shao-Sung Huang, Wan-Leong Chan, Shing-Jong Lin, Jaw-Wen Chen, Hsin-Bang Leu.

**Writing – original draft:** Tsung-Ying Tsai.

**Writing – review & editing:** Pai-Feng Hsu, Yuan-Jen Wang, Yaw-Zon Ding, Teh-Ling Liou, Ying-Wen Wang, Shao-Sung Huang, Wan-Leong Chan, Shing-Jong Lin, Jaw-Wen Chen, Hsin-Bang Leu.

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
