## [Decision Letter · Decision Letter 0]

2 Jul 2020

PONE-D-20-13875

Factor analysis for the clustering of cardiometabolic risk factors and sedentary behavior

PLOS ONE

Dear Dr. Leu,

Thank you for submitting your manuscript to PLOS ONE. After careful consideration, we feel that it has merit but does not fully meet PLOS ONE’s publication criteria as it currently stands. Therefore, we invite you to submit a revised version of the manuscript that addresses the points raised during the review process.

We look forward to receiving your revised manuscript.

Kind regards,

Mauro Lombardo

Academic Editor

PLOS ONE

Journal Requirements:

2. Please provide additional details regarding participant consent. In the ethics statement in the Methods and online submission information, please ensure that you have specified what type of consent you obtained (for instance, written or verbal, and if verbal, how it was documented and witnessed).

3.We note that you have indicated that data from this study are available upon request. PLOS only allows data to be available upon request if there are legal or ethical restrictions on sharing data publicly. For information on unacceptable data access restrictions, please see http://journals.plos.org/plosone/s/data-availability#loc-unacceptable-data-access-restrictions.

4.Thank you for stating the following financial disclosure:

 [The funders had no role in study design, data collection and analysis, decision to publish, or preparation of the manuscript.]

6.Please amend your list of authors on the manuscript to ensure that each author is linked to an affiliation. Authors’ affiliations should reflect the institution where the work was done (if authors moved subsequently, you can also list the new affiliation stating “current affiliation:….” as necessary).

<h1>****</h1>

Reviewers' comments:

Reviewer's Responses to Questions

**Comments to the Author**

1. Is the manuscript technically sound, and do the data support the conclusions?

Reviewer #1: Yes

Reviewer #2: Partly

2. Has the statistical analysis been performed appropriately and rigorously? 

Reviewer #1: No

Reviewer #2: Yes

3. Have the authors made all data underlying the findings in their manuscript fully available?

Reviewer #1: Yes

Reviewer #2: Yes

4. Is the manuscript presented in an intelligible fashion and written in standard English?

Reviewer #1: Yes

Reviewer #2: Yes

5. Review Comments to the Author

Reviewer #1: This paper describes results of analysis to explore the clustering pattern of CVD risk factors in a Taiwanese study population. Although the paper is very well written, I have a few concerns which I explain below.

In the introduction the authors state that few studies have investigated the clustering relationship between sedentary behavior and other risk factors, and there is insufficient information to explain the variances of metabolic abnormalities observed. However, the authors did not give any references and I doubt whether this is actually true. I believe that there is a battery of studies showing that sedentary behavior clusters with factors of the metabolic syndrome? The reasons for why you did this study are not clear to me. Please explain and rephrase this part so that limitations of previous studies can be explained in further detail.

Why did respondents receive a comprehensive health examination at the Healthcare Center of Taipei Veterans 96 General Hospital from February 2015 to July 2019? In the results section it says that respondents are healthy, but why did they receive a health examination? Did they visit a specialist in the hospital? If so, they don’t seem to be healthy. What were further inclusion criteria and what were the response rates to this questionnaire?

It is not clear to me why respondents were divided in a training and validation group. Please explain. The numbers seem relatively small anyway, why not studying the group as a whole? Furthermore, to me, it seems more logic to perform another factor analysis on another study population (with different characteristics), rather than a sample of the same study population with similar characteristics.

Have the authors considered the method of machine learning (e.g. random forest)?

The result that there was no correlation between MET or sitting with lipid profiles, but that respondents with low physical activity have unfavorable lipid profiles and higher levels of HsCrp puzzles me. To be able to interpret this correctly, corrections for potential confounders, such as age and sex, etc, should be applied at the least.

The findings that adiposity is linked to inflammation and sedentary behavior is clustered with physical activity are not new to me. The manuscript lacks novelty. Furthermore, no explanation is given for that the explained variance for physical activity is relatively low.

A huge limitation of this study which is currently missing in this section is the cross-sectional data.

Reviewer #2: Leu et al. investigates cluster of CV risk factors in an Asian population. Authors identify five clusters with sedentary behavior explaining the lowest variance. An interesting finding was that inflammation biomarker HsCRP was clustered with only adiposity factors and not with other cardiometabolic risk factors.

Comments

-Authors imply they have looked at clusters of new CV risk factors. However, CRP and physical inactivity are not new CV risk factors.

-Authors mention in the method section they have data on haematological parameters (e.g., haemoglobin) which are emerging as novel risk factors for CVD. Why did authors not include these parameters in their analysis? Same questions applied also to liver enzymes and alcohol intake.

-It is not clear from method section whether participants in the current study were free of chronic diseases, including dyslipidaemia, hypertension and CVD. Were participants taking any medications? If yes, how was that taken into account in the analysis?

-How do authors explain the low correlation between sitting time and total physical activity? Wouldn’t you expect a higher correlation between the two factors? How this correlation observed in the current study contrasts previous findings? Maybe the questionnaires used in this study to capture both physical activity and inactivity were not good enough?

-It would be of interest to explore age and sex-stratified analysis, and see whether the clusters differ by age groups and sex. The paper would bring more novelty if the analysis were since the beginning stratified by sex, considering the sex gap in CVD.

6. PLOS authors have the option to publish the peer review history of their article (what does this mean?). If published, this will include your full peer review and any attached files.

Reviewer #1: **Yes: **Gerrie-Cor Herber

Reviewer #2: No

---

## [Author Response · Author response to Decision Letter 0]

21 Aug 2020

Response to reviewer 1

We appreciate your detailed comments, which have helped to improve our manuscript. Our responses to your comments are presented below and the passages that were added to the text are underlined in the revised manuscript.

Reply to the comments

1. Regarding the comments: "In the introduction the authors state that few studies have investigated the clustering relationship between sedentary behavior and other risk factors, and there is insufficient information to explain the variances of metabolic abnormalities observed. However, the authors did not give any references and I doubt whether this is actually true. I believe that there is a battery of studies showing that sedentary behavior clusters with factors of the metabolic syndrome? The reasons for why you did this study are not clear to me. Please explain and rephrase this part so that limitations of previous studies can be explained in further detail.".

Reply: Thank you for your comments and we apologize that our manuscript was unclear. We agree that studies have investigated clustering factor components with or without physical activity in metabolic syndrome [1–5]. However, few studies have considered inflammatory biomarkers among the clustering risk factors. In addition, although avoiding sedentary behavior has been suggested by international guidelines, there is no clear consensus on the definition of sedentary behavior [6]. Our current study used the IPAQ questionnaire, which reported physical activity volume (MET) and sitting time information at the same time, providing the opportunity to investigate the impact of sitting time/MET as well as inflammatory biomarkers in the same factor analysis model. The advantage of performing exploratory factor analysis rather than simply performing a linear correlation as that factor analysis can reveal the correlation and the underlying factor structure of a set of factors without imposing a preconceived structure on the outcome. Thus, we believe that our study design is valuable compared to the previous studies. We have revised the introduction of the manuscript to clarify the aim of our study. (page 4, lines70 to page 4, line 73, and page 5, line 77 to page 5, line 81)

Reference

1. HealyGN, MatthewsCE, DunstanDW, WinklerEAH, OwenN. Sedentary time and cardio-metabolic biomarkers in US adults: NHANES 200306. Eur Heart J. 2011;32: 590–597. doi:10.1093/eurheartj/ehq451

2. EdwardsonCL, GorelyT, DaviesMJ, GrayLJ, KhuntiK, WilmotEG, et al. Association of Sedentary Behaviour with Metabolic Syndrome: A Meta-Analysis. O’ConnorKA, editor. PLoS One. 2012;7: e34916. doi:10.1371/journal.pone.0034916

3. SungwachaJN, TylerJ, Longo-MbenzaB, Lasi On’KinJBK, GombetT, ErasmusRT. Assessing clustering of metabolic syndrome components available at primary care for Bantu Africans using factor analysis in the general population. BMC Res Notes. 2013;6: 228. doi:10.1186/1756-0500-6-228

4. EsteghamatiA, ZandiehA, KhalilzadehO, MortezaA, MeysamieA, NakhjavaniM, et al. Clustering of leptin and physical activity with components of metabolic syndrome in Iranian population: An exploratory factor analysis. Endocrine. 2010;38: 206–213. doi:10.1007/s12020-010-9374-9

5. KelishadiR, ArdalanG, AdeliK, MotaghianM, MajdzadehR, Mahmood-ArabiMS, et al. Factor Analysis of Cardiovascular Risk Clustering in Pediatric Metabolic Syndrome: CASPIAN Study. Ann Nutr Metab. 2007;51: 208–215. doi:10.1159/000104139

6. TremblayMS, AubertS, BarnesJD, SaundersTJ, CarsonV, Latimer-CheungAE, et al. Sedentary Behavior Research Network (SBRN) – Terminology Consensus Project process and outcome. Int J Behav Nutr Phys Act. 2017;14: 75. doi:10.1186/s12966-017-0525-8

2. Regarding the comments: "Why did respondents receive a comprehensive health examination at the Healthcare Center of Taipei Veterans 96 General Hospital from February 2015 to July 2019? In the results section it says that respondents are healthy, but why did they receive a health examination? Did they visit a specialist in the hospital? If so, they don’t seem to be healthy. What were further inclusion criteria and what were the response rates to this questionnaire?"

Reply: We appreciate this helpful comment and apologize that this information was unclear. Our study derived data from the healthcare center of Taipei Veterans general hospital, which provides elective, self-paid health examination services to all individuals. In Taiwan, the healthcare service is covered by the national health insurance system (NHI), which guarantees nearly full reimbursement should a patient develop any medical illness or discomfort that requires a clinic visit or hospital evaluation [7]. However, for those without significant symptoms or illness, but who would like an examination in advance to identify undetected conditions for primary prevention purposes, the cost of the check-up is not covered by the NHI system, and the patient pays for it. We believe that data from our study population could provide vital information on primary prevention from the general population perspective. As seen in Table 1 in the manuscript, the prevalence of comorbidities in our population was quite low. Thus, their data reflect the healthy general population of Taiwan. For the response rate, there were 15,636 patients who had received check-up services during the study period, and 5606 (35.9%) subjects agreed to participate in our study. All participants completed the questionnaires. To avoid ambiguity, we added a more detailed description in the Methods section (page 6, lines 100 and page 7, line 110).

Reference

7. HsiehC-Y, SuC-C, ShaoS-C, SungS-F, LinS-J, Yang KaoY-H, et al. 

Taiwan’s National Health Insurance Research Database: past and future

. Clin Epidemiol. 2019;Volume 11: 349–358. doi:10.2147/CLEP.S196293

3. Regarding the comments: "It is not clear to me why respondents were divided in a training and validation group. Please explain. The numbers seem relatively small anyway, why not studying the group as a whole? Furthermore, to me, it seems more logic to perform another factor analysis on another study population (with different characteristics), rather than a sample of the same study population with similar characteristics”.

Reply: Thank you for your comments and we apologize that the rationale for performing a validation analysis was unclear in our manuscript. Our study subjects were relative healthy without major diseases. They underwent a detailed check-up for primary prevention purposes. We designed our study this way so that we could analyze the different variable components of variance that were not affected by underlying disease or medications. Because it is difficult to find another population with the same low risk who had also completed the questionnaires as well as a biochemical study including CRP, we consider that it was reasonable to divide the whole population into two groups, one for training and one for validation to avoid overfitting the developed model. In statistics for data mining, it is a common method that training data is often a subset of the data set, and the test set is a subset of the test-trained model. Similar analyses have been reported previously. For example, in Hravnak’s machine learning algorithms data were divided into Block 1 for the ML training/cross-validation set and Block 2 for the test set [8]. Furthermore, Goodman et al. used factor analysis for the cardiovascular clustering risk, and they selected 20% of the cases as an exploratory sample and the remaining 80% cases were used as a validation sample [9]. Because our current study aimed to investigate the cluster risk including inflammation makers and sedentary behavior information, it is reasonable to test and validate sample using the same number of variables. We have modified the Methods section of the manuscript to describe our design. (page 11, lines 180 to page 120, line 195).

Reference

8. HravnakM, ChenL, DubrawskiA, BoseE, ClermontG, PinskyMR. Real alerts and artifact classification in archived multi-signal vital sign monitoring data: implications for mining big data. J Clin Monit Comput. 2016;30: 875–888. doi:10.1007/s10877-015-9788-2

9. GoodmanE, DolanLM, MorrisonJA, DanielsSR. Factor analysis of clustered cardiovascular risks in adolescence: obesity is the predominant correlate of risk among youth. Circulation. 2005;111: 1970–7. doi:10.1161/01.CIR.0000161957.34198.2B

4. Regarding the comments: "Have the authors considered the method of machine learning (e.g. random forest)?”

Reply: We appreciate your instructive comment about using machine learning on our data. The current study is one of the initial studies that was performed using from the VGH-HEATHCARE database, and it aimed to dissect the complexity of clustering risk factors in the general population. The machine learning method, such as the random forest technique, is a powerful tool for building risk prediction models in both cross-sectional and cohort studies [10,11]. Because the VGH-HEATHCARE study is a prospective cohort study that still ongoing, machine learning studies will be conducted in the future to continue building the risk prediction model after a long-term outcome is available in the future. Thank you for your invaluable suggestion. 

Reference

10. KanervaN, KonttoJ, ErkkolaM, NevalainenJ, MännistöS. Suitability of random forest analysis for epidemiological research: Exploring sociodemographic and lifestyle-related risk factors of overweight in a cross-sectional design. Scand J Public Health. 2018;46: 557–564. doi:10.1177/1403494817736944

11. WengSF, RepsJ, KaiJ, GaribaldiJM, QureshiN. Can machine-learning improve cardiovascular risk prediction using routine clinical data? LiuB, editor. PLoS One. 2017;12: e0174944. doi:10.1371/journal.pone.0174944

5. Regarding the comments: "The result that there was no correlation between MET or sitting with lipid profiles, but that respondents with low physical activity have unfavorable lipid profiles and higher levels of HsCrp puzzles me. To be able to interpret this correctly, corrections for potential confounders, such as age and sex, etc, should be applied at the least”.

Reply: We appreciate your insightful comment, and we apologize that our results were unclear. As shown in Table 2 of the manuscript, the MET value was significantly negatively correlated with waist circumference, blood pressure, cholesterol, triglyceride, LDL, and sitting time, but positively correlated with HDL. We have further categorized subjects into low, moderate, and high physical activity group, and we found that subjects with low activity have an unfavorable lipid profile and higher baseline inflammatory marker levels, supporting the connection between higher cardiovascular risk and low physical activity. Although sitting time is negatively correlated with MET values, the correlation between lipid profiles only existed for HDL. This is not surprising because sitting time is only one factor among the definition of sedentary behavior and total MET activity estimation was generated using other activity information. We re-analyzed the correlation, taking into consideration gender and age according to your suggestion. (supplemental Table 1, 2, 3, and 4) Thank you again for your suggestion 

We have modified the Results of the manuscript. (page 13, lines 206 to page 14, line 219).

6. Regarding the comments:” The findings that adiposity is linked to inflammation and sedentary behavior is clustered with physical activity are not new to me. The manuscript lacks novelty. Furthermore, no explanation is given for that the explained variance for physical activity is relatively low.”

Reply: We appreciate your comment that pointed out the limitation of our study. The findings that adiposity is linked with inflammation and sedentary behavior is clustered with physical activity are not new. The novelty of our study is as follows: (1) we are the first to explore the clustering structure of CV risk factors and physical activity together; and (2) we demonstrated that the physical activity factors do not cluster with traditional CV risk factors. Our result suggests that physical inactivity may exert its effect on cardiovascular disease in an independent and unique way. This result may prompt future researchers to explore the possible pathophysiologic mechanism behind the independent effect of the physical activity level. The low variance of physical activity reflects the low physical activity in our healthy subjects. Table 3 in our manuscript shows that nearly 43% of our participants belong to the low activity categories, indicating a relatively inadequate physical activity level. Because our study population was enrolled from a self-pay check-up population, there might be a selection bias that this population may have higher socioeconomic status, but lower daily activity. However, the high prevalence of low physical activity is not unique to our population because other studies have demonstrated similar low physical activity levels. A recent global heath observatory survey showed that 23% of men and 32% of women ≥18 years of age were insufficiently physically active. Over the past 15 years, the levels of insufficient activity did not improve (28.5% in 2001; 27.5% in 2016). The WHO regions of the Americas (39%) and the Eastern Mediterranean Region (35%) had the highest prevalence of insufficient physical activity [12,13]. 

Reference

12. WHO | Prevalence of insufficient physical activity. WHO. 2018.

13. CDC maps show variance in adult physical activity levels by state, territory | AHA News. [cited 11 Aug 2020]. Available: https://www.healthycommunities.org/news/headline/2020-01-22-cdc-maps-show-variance-adult-physical-activity-levels-state-territory

7. Regarding the comments:”A huge limitation of this study which is currently missing in this section is the cross-sectional data”

Reply: Thank you and we agree with your opinion that our study is a cross-sectional study that lacks long-term outcome information. Because the VGH-HEALTHCARE study is still ongoing, we can only wait for the long-term outcome data in the future. We have mentioned this limitation of cross-sectional data with the study limitations, and we also revised the title, adding “a cross-sectional study”. Thank you for your suggestion. (page 21, lines 336 to page 21 line 338).

Responses to reviewer 2

We appreciate your helpful comments. Our responses are presented below, and the relevant passages have been incorporated into the revised manuscript.

1. Regarding the comments: " Authors imply they have looked at clusters of new CV risk factors. However, CRP and physical inactivity are not new CV risk factors. “

Reply: We appreciate your opinion that CRP is not a new marker. Our paper aimed to dissect various cardiometabolic risk factors that are easy to obtain simultaneously in the general population. We apologize that this was unclear in our manuscript, and it has been revised accordingly.

2. Regarding the comments: “Authors mention in the method section they have data on haematological parameters (e.g., haemoglobin) which are emerging as novel risk factors for CVD. Why did authors not include these parameters in their analysis? Same questions applied also to liver enzymes and alcohol intake."

Reply: Thank you for this insightful comment. Our current study enrolled subjects who underwent an annual physical check-up at VGH. Although the hemoglobin value and liver function are considered to be new cardiovascular markers, there is little evidence supporting their role in the general population compared with the factors in our studies. Therefore, we only enrolled the most important markers. In addition, the information about alcohol intake in our study was incomplete. We do not have information on the type of alcohol (e.g. wine, beer, or whisky) or the amount of consumption. Therefore, these factors were not included in the final analysis. We have added this as a study limitation. Thank you for your comments. (page 21, lines 339 to page 21 line 342).

3. Regarding the comments: " It is not clear from method section whether participants in the current study were free of chronic diseases, including dyslipidaemia, hypertension and CVD. Were participants taking any medications? If yes, how was that taken into account in the analysis?"

Reply: We appreciate this helpful comment and apologize for the lack of clarity in presenting our results. Our study derived data from the Taipei Veterans general hospital healthcare center that provides a health examination service to all individuals. However, for most of the population and most aspects of healthcare that is covered by national health insurance, patients with a particular complaint would usually have no difficulty finding a specialist appointment [7]. Our health examination services were almost exclusively used by health-conscious individuals who simply wished to receive comprehensive check-ups or who were concerned about having an undetected health issue. As seen in our data, the number of comorbidities in our population was quite low. Thus, they fit our initial description of healthy community-dwelling individuals. To avoid ambiguity, we have added a more detailed description to the Methods section. (page 6, lines 100 and page 7, line 110).

Reference

7. HsiehC-Y, SuC-C, ShaoS-C, SungS-F, LinS-J, Yang KaoY-H, et al. 

Taiwan’s National Health Insurance Research Database: past and future

. Clin Epidemiol. 2019;Volume 11: 349–358. doi:10.2147/CLEP.S196293

Regarding the comments: "How do authors explain the low correlation between sitting time and total physical activity? Wouldn’t you expect a higher correlation between the two factors? How this correlation observed in the current study contrasts previous findings? Maybe the questionnaires used in this study to capture both physical activity and inactivity were not good enough?”

Reply: We appreciate this comment. Traditionally, sedentary behavior and total physical activity are believed to be independent risk factors for cardiovascular events [14,15]. However, the association between physical activity and sedentary behavior is not consistent. Some studies reported no significant association between the two while others showed that there was some association [16,17]. This discrepancy can be explained by the fact that a person can be both sedentary and physically active (the Active Couch Potato phenomenon, for example, would be an office worker who jogs or bikes to and from work, but who then sits all day at a desk and spends several hours watching TV in the evening) [18]. In this study, we used the IPAQ questionnaire, which is a widely validated questionnaire that is recommended by the WHO, to evaluate the association between sedentary behavior and weekly physical activity [19,20]. We believe that the low but still significant correlation between sitting time is consistent with previous studies, and it can be explained by the fact that non-sedentary physical activity in the general population is quite variable.

 Reference

14. PandeyA, SalahuddinU, GargS, AyersC, KulinskiJ, AnandV, et al. Continuous dose-response association between sedentary time and risk for cardiovascular disease a meta-analysis. JAMA Cardiol. 2016;1: 575–583. doi:10.1001/jamacardio.2016.1567

15. EkelundU, BaumanA, LeeIM. Effects of early physical exercise on later health – Authors’ reply. The Lancet. Lancet Publishing Group; 2017. p. 801. doi:10.1016/S0140-6736(17)30506-8

16. SilfeeV, LemonS, LoraV, RosalM. Sedentary behavior and cardiovascular disease risk factors among latino adults. J Health Care Poor Underserved. 2017;28: 798–811. doi:10.1353/hpu.2017.0075

17. ChomistekAK, MansonJE, StefanickML, LuB, Sands-LincolnM, GoingSB, et al. Relationship of sedentary behavior and physical activity to incident cardiovascular disease: Results from the women’s health initiative. J Am Coll Cardiol. 2013;61: 2346–2354. doi:10.1016/j.jacc.2013.03.031

18. OwenN, HealyGN, MatthewsCE, DunstanDW. Too much sitting: The population health science of sedentary behavior. Exerc Sport Sci Rev. 2010;38: 105–113. doi:10.1097/JES.0b013e3181e373a2

19. CraigCL, MarshallAL, SjöströmM, BaumanAE, BoothML, AinsworthBE, et al. International physical activity questionnaire: 12-Country reliability and validity. Med Sci Sports Exerc. 2003;35: 1381–1395. doi:10.1249/01.MSS.0000078924.61453.FB

20. BoonRM, HamlinMJ, SteelGD, RossJJ. Validation of the New Zealand physical activity questionnaire (NZPAQ-LF) and the international physical activity questionnaire (IPAQ-LF) with accelerometry. Br J Sports Med. 2010;44: 741–746. doi:10.1136/bjsm.2008.052167

4. Regarding the comments: "It would be of interest to explore age and sex-stratified analysis, and see whether the clusters differ by age groups and sex. The paper would bring more novelty if the analysis were since the beginning stratified by sex, considering the sex gap in CVD”.

Reply: We appreciate this important comment. We had not considered age and sex-stratified analysis, but this is an important direction to explore. Thus, we performed an age and sex-stratified analysis as you suggested. The results showed a similar clustering pattern of risk factors in both gender and age groups (Supplemental Tables 1, 2, 3, and 4). This recommendation is greatly appreciated because the addition of sex and age-stratified results strengthen our study. We have mentioned the results of this analysis in our revised manuscript. (page 16, line 244 to page 16, line 247)

---

## [Decision Letter · Decision Letter 1]

28 Sep 2020

PONE-D-20-13875R1

Factor analysis for the clustering of cardiometabolic risk factors and sedentary behavior, a cross-sectional study.

PLOS ONE

Dear Dr. Leu,

Thank you for submitting your manuscript to PLOS ONE. After careful consideration, we feel that it has merit but does not fully meet PLOS ONE’s publication criteria as it currently stands. Therefore, we invite you to submit a revised version of the manuscript that addresses the points raised during the review process.

We look forward to receiving your revised manuscript.

Kind regards,

Mauro Lombardo

Academic Editor

PLOS ONE

Reviewers' comments:

Reviewer's Responses to Questions

**Comments to the Author**

1. If the authors have adequately addressed your comments raised in a previous round of review and you feel that this manuscript is now acceptable for publication, you may indicate that here to bypass the “Comments to the Author” section, enter your conflict of interest statement in the “Confidential to Editor” section, and submit your "Accept" recommendation.

Reviewer #2: All comments have been addressed

Reviewer #3: (No Response)

Reviewer #4: All comments have been addressed

2. Is the manuscript technically sound, and do the data support the conclusions?

Reviewer #2: Yes

Reviewer #3: Partly

Reviewer #4: Yes

3. Has the statistical analysis been performed appropriately and rigorously? 

Reviewer #2: Yes

Reviewer #3: (No Response)

Reviewer #4: Yes

4. Have the authors made all data underlying the findings in their manuscript fully available?

Reviewer #2: Yes

Reviewer #3: Yes

Reviewer #4: Yes

5. Is the manuscript presented in an intelligible fashion and written in standard English?

Reviewer #2: Yes

Reviewer #3: (No Response)

Reviewer #4: Yes

6. Review Comments to the Author

Reviewer #2: Authors have successfully addressed my suggestions; the analysis on sex differences is of interest, despite no differences between men and women were found. I do not have further comments

Reviewer #3: More work is needed from the author, particularly in the method section, to make the manuscript scientifically suitable for publication.

Reviewer #4: Dear editor in chief

Thank you for inviting me to review the above-referenced paper. This research paper by Tsung-Ying Tsaiet al., aims to investigate the clustering relationship of sedentary behavior, cardiometabolic components, and inflammatory biomarkers among 5606 adults in Taiwan. This was investigated using an exploratory factor analysis. They found five cardiometabolic risk factors clusters: the adiposity factor (waist circumference, BMI, TG, HDL, and UA), the blood pressure factor (SBP and DBP), the lipid factor (TC and LDL), the glucose factor (fasting glucose and HbA1C), and the physical activity factor. Inflammation biomarker was clustered with adiposity factors, while physical inactivity and sedentary behavior were clustered with other factors.

I think the manuscript can be accepted for publication after a minor revision:

1. In "Statistical analysis" section, it's written" categorical variables were expressed as the mean ± 95% confidence interval but it was expressed as number and percentage.

2. In the discussion section, It's written "Table 3 shows the baseline characteristics of all subjects according to physical activity". and "Indeed, as shown in Table 3, patients with low physical activity have multiple factors at the same time, making it difficult to

evaluate the importance and correlation of individual risk factors". Here there are two remarks: first, the description of the results of the table are described in the result section and not the discussion section; Second, table 3 shows the Factor analysis of the training group and Table 5 shows the baseline characteristics of all subjects according to physical activity.

7. PLOS authors have the option to publish the peer review history of their article (what does this mean?). If published, this will include your full peer review and any attached files.

Reviewer #2: No

Reviewer #3: No

Reviewer #4: No

---

## [Author Response · Author response to Decision Letter 1]

19 Oct 2020

Point to point response to reviewers 

Response to reviewer 4

Regarding the comments: "In "Statistical analysis" section, it's written" categorical variables were expressed as the mean ± 95% confidence interval but it was expressed as number and percentage”.

Reply: Thank you for your comments, we apologize for this incorrect statement. We have revised the statement in the method section to fit the presentation in the results section. (page 10 lines 10)

Regarding the comments: "In the discussion section, It's written "Table 3 shows the baseline characteristics of all subjects according to physical activity". and "Indeed, as shown in Table 3, patients with low physical activity have multiple factors at the same time, making it difficult to evaluate the importance and correlation of individual risk factors". Here there are two remarks: first, the description of the results of the table are described in the result section and not the discussion section; Second, table 3 shows the Factor analysis of the training group and Table 5 shows the baseline characteristics of all subjects according to physical activity.”

Reply: Thank you for your comments, we apologize for the mislabeling. We have moved the statement regarding table 3 to the discussion section and labeled the statements regarding the tables correctly in the revised manuscript. 

(page 21 lines 4 to 6; page 34)

 

Response to reviewer 1

1. Reviewer 1 wondered that there is a battery of studies showing that sedentary behavior clusters with factors of the metabolic syndrome, and claimed that knowledge gaps were not clearly explained.

a. I agree with the reviewer.

b. Authors’ reply to this comment was not exactly relevant to the question.

c. In Response to this comment, the authors failed to appropriately revising the manuscript. They mentioned that “There has been is a battery of studies demonstrating the association between sedentary and tradition CVD risk factors”; however, the authors did not explain that association, and they did not give an example from a study examined such association to support the introductory statement (i.e., Ln 68-70). The authors were not clear if the phathophysiology of the association between sedentary behavior and CVD was previously explained or not. In addition, the authors in Ln 75-78 contrary the introductory statement where they stated “few studies”. Finally, the authors failed to state knowledge gaps, and then justify their study.

Reply: We appreciate this comment from reviewer 1 and apologize that our manuscript was not fully respond to the reviewer’s requirement. There are many studies reporting the association between sedentary behavior and various cardiovascular risk factors. For example, in the landmark NHANES 2003-2006 study, total sedentary time was detrimentally associated with several biomarkers including waist circumference, HDL-cholesterol, C-reactive protein, triglycerides, insulin, and insulin resistance. Breaks from long sitting period, independent of sedentary time, were beneficially associated with waist circumference, C-reactive protein, and fasting plasma glucose. However, while this study only demonstrated the close relationship between sedentary behavior and inflammation, other important risk factors such as total cholesterol, UA, HbA1C, were not evaluated in that study. Moreover, only a part of the study population has serum glucose measurement in this study. [8] In a later meta-analysis of more than 20000 subjects, Edwardson et al. demonstrated that patients with longer sedentary time have greater odds of having metabolic syndrome. However, metabolic syndrome represented a clinical syndrome with heterogenous characteristics and it is impossible to delineate the association between sedentary behavior and a particular component of metabolic syndrome from these evidences.[9] Thus, although sedentary behavior has been shown to be associated with many cardiovascular risk factors, the more detailed clustering relationship was unknown. Factor analysis has a unique advantage revealing the clustering structure of various risk factors. However, previous sedentary behavior rarely included sedentary behavior while those that did do not include major cardiovascular risk factors. Hence our study has the advantage to analyze the clustering structure of most used cardiovascular risk factors and sedentary behavior. We have revised our manuscript and added above descriptions in the section of Introduction. In addition, we have also added a statement about the pathophysiologic implication of sedentary behavior that sedentary behavior linked to reduced triglyceride metabolism, insufficient antioxidant production, and glucose intolerance in several animal studies. (page 5, lines 13 to page 6 line 13). 

References

8. HealyGN, MatthewsCE, DunstanDW, WinklerEAH, OwenN. Sedentary time and cardio-metabolic biomarkers in US adults: NHANES 200306. Eur Heart J. 2011;32: 590–597. doi:10.1093/eurheartj/ehq451

9. EdwardsonCL, GorelyT, DaviesMJ, GrayLJ, KhuntiK, WilmotEG, et al. Association of Sedentary Behaviour with Metabolic Syndrome: A Meta-Analysis. O’ConnorKA, editor. PLoS One. 2012;7: e34916. doi:10.1371/journal.pone.0034916

2. Comment 2 of the reviewer 1, as well as comments 2 and 3 of the reviewer 2 was mainly about inclusion and exclusion criteria. 

a. Authors’ reply to this comment was not exactly relevant to the question. In general, the authors do not need to emphasize the self-paid issue. Instead, they need to clearly define the study population and study patients. In addition, they must clearly state inclusion and exclusion criteria, which were overlooked in the revised manuscript. 

b. The revised manuscript (i.e., Ln 98-108) included not needed information, while overlooked important information such as the reason for selecting this healthcare center. Taking into account that self-paid reason for selecting this center is not relevant, unless this issue is matter in the analysis.

Reply: We appreciate this comment from reviewer 1. We have removed some redundant description about the self-paid reason and added the following statement to describe our inclusion and exclusion criteria. “We included patients without significant symptoms or illness and excluded those who refused to participate, whose exam revealed an acute illness, or had a chronic condition that require regular follow up such as active cancer, heart failure, coronary artery disease or stroke. “(page 8, lines 6 to line 10 )

3. Reviewer 1 asked the authors to explain the training and validation groups, and questioned the study population and study patients. 

a. I agree with the reviewer. 

b. Authors’ response is not exactly relevant.

c. The authors need to define both training and validation groups in the current study, and the criteria used for selecting the subjects to such group.

d. Again, the authors need to clearly state inclusion and exclusion criteria.

Reply: We appreciate this instructive comment from reviewer 1. The training group and the validation group were randomly selected from the total study population. We have described the allocation process in the revised manuscript. 

(page 13, lines 2 to line 3) 

4. Reviewer 1 asked if the authors considered the method of machine learning in their methodology. 

a. Authors’ response is inconvenience. 

b. In the revised manuscript, Ln 181-185 was not clear.

c. In the revised manuscript, authors explained some methodology for machine learning, whereas the current study considered factor analysis. Therefore, Ln 186-187 is not relevant to the current study method.

d. In the revised manuscript, Ln 188-193 neither relevant nor clear. The authors should justify classifying such study subjects as training group and others as validation group.

Reply: We appreciate this comment from reviewer. We have added a short description about not considering machine learning methods in our study limitation. We will use machine learning method for the analysis of outcome data after the follow-ups of the VGH-HEALTHCARE study is completed. For the current study, we did not consider using machine learning in our methodology for two reasons. First, using machine learning technology for factor analysis has not been widely accepted because of lack of long-term outcome information. Second, our study purpose would like to investigate the percentage of variance of clustering cardiometabolic risk factors and the association between sedentary behavior and other associated risk factors. 

 (Page 23, line 5 to line 13)

5. In the comment 5 of the reviewer 1 and comment 4 of the reviewer 2, the analysis should consider the potential confounding effect of the age and sex. 

• The authors re-analyzed that correlation and provided supplemental tables.

Reply: We appreciate this comment from the reviewer.

6. Reviewer 1 in the comment 6 claimed that the current manuscript lacks the novelty and explanation for relatively low variance for physical activity.

a. The authors explained in their reply this issue; however, it was not clear in the revised manuscript. They need to clarify and state it in the revised manuscript.

b. In the reply, the authors mentioned that “A recent global heath observatory survey showed that 23% of men and 32% of women ≥18 years of age were insufficiently physically active. Over the past 15 years, the levels of insufficient activity did not improve (28.5% in 2001; 27.5% in 2016).” Does this apply to your country?

Add novelty in revised manuscript 

Reply: Thank you for your comment. As stated in our previous response, this study is the first to investigate the clustering relationship of a comprehensive array of cardiometabolic factors, including systemic inflammation, and sedentary information simultaneously in the general population of Taiwan. We also demonstrated that the physical activity factors do not cluster with traditional CV risk factors. Our results suggest that physical inactivity may exert its effect on cardiovascular disease in an independent and unique way. This result may prompt future researchers to explore the possible pathophysiologic mechanism behind the independent effect of the physical activity level. We have added the novelty of this study in the revised manuscript. In response to your second comment, inadequate physical activity was observed in the Taiwanese population as well.[42] Although our data is not a good representative of all Taiwanese people, the percentage is reasonable for the Taiwanese general population. There may be a selection bias, that we cannot avoid selecting subjects with better higher social economic status whose sedentary behavior may be different from other population. (page 17, lines 11 to line 17)

References:

42. WuX, TsaiSP, TsaoCK, ChiuML, TsaiMK, LuPJ, et al. Cohort Profile: The Taiwan MJ Cohort: Half a million Chinese with repeated health surveillance data. Int J Epidemiol. 2017;46: 1744-1744g. doi:10.1093/ije/dyw282

7. Reviewer 2 wondered "How do authors explain the low correlation between sitting time and total physical activity?”

a. I agree with the reviewer.

b. The authors need to make the revised as clear as their reply to this comment.

Reply: In our previous response to reviewers, we have addressed this issue by stating that traditionally, sedentary behavior and total physical activity are believed to be independent risk factors for cardiovascular events [6,7]. However, the association between physical activity and sedentary behavior is not consistent. Some studies reported no significant association between the two while others showed that there was some association [43]. This discrepancy can be explained by the fact that a person can be both sedentary and physically active (the Active Couch Potato phenomenon, describes someone who meets the recommendations for physical activity but still sits around for long periods of the day.) [44]. In this study, we used the IPAQ questionnaire, which is a widely validated questionnaire that is recommended by the WHO, to evaluate the association between sedentary behavior and weekly physical activity [45,46]. We believe that the low but still significant correlation between sitting time is consistent with previous studies, and it can be explained by the fact that non-sedentary physical activity in the general population is quite variable. We believe our original description should be sufficient. 

 References:

6. PandeyA, SalahuddinU, GargS, AyersC, KulinskiJ, AnandV, et al. Continuous dose-response association between sedentary time and risk for cardiovascular disease a meta-analysis. JAMA Cardiol. 2016;1: 575–583. doi:10.1001/jamacardio.2016.1567

7. EkelundU, BaumanA, LeeIM. Effects of early physical exercise on later health – Authors’ reply. The Lancet. Lancet Publishing Group; 2017. p. 801. doi:10.1016/S0140-6736(17)30506-8

43. SilfeeV, LemonS, LoraV, RosalM. Sedentary behavior and cardiovascular disease risk factors among latino adults. J Health Care Poor Underserved. 2017;28: 798–811. doi:10.1353/hpu.2017.0075

44. OwenN, HealyGN, MatthewsCE, DunstanDW. Too much sitting: The population health science of sedentary behavior. Exerc Sport Sci Rev. 2010;38: 105–113. doi:10.1097/JES.0b013e3181e373a2

45. CraigCL, MarshallAL, SjöströmM, BaumanAE, BoothML, AinsworthBE, et al. International physical activity questionnaire: 12-Country reliability and validity. Med Sci Sports Exerc. 2003;35: 1381–1395. doi:10.1249/01.MSS.0000078924.61453.FB

46. BoonRM, HamlinMJ, SteelGD, RossJJ. Validation of the New Zealand physical activity questionnaire (NZPAQ-LF) and the international physical activity questionnaire (IPAQ-LF) with accelerometry. Br J Sports Med. 2010;44: 741–746. doi:10.1136/bjsm.2008.052167

8. Reviewer 1 claimed a huge limitation in this study.

• The authors revised the manuscript, and added a section for study limitations.

Reply: We appreciate your comment.

---

## [Decision Letter · Decision Letter 2]

2 Nov 2020

Factor analysis for the clustering of cardiometabolic risk factors and sedentary behavior, a cross-sectional study.

PONE-D-20-13875R2

Dear Dr. Leu,

We’re pleased to inform you that your manuscript has been judged scientifically suitable for publication and will be formally accepted for publication once it meets all outstanding technical requirements.

Kind regards,

Mauro Lombardo

Academic Editor

PLOS ONE

Additional Editor Comments (optional):

Reviewers' comments:

Reviewer's Responses to Questions

**Comments to the Author**

1. If the authors have adequately addressed your comments raised in a previous round of review and you feel that this manuscript is now acceptable for publication, you may indicate that here to bypass the “Comments to the Author” section, enter your conflict of interest statement in the “Confidential to Editor” section, and submit your "Accept" recommendation.

Reviewer #3: All comments have been addressed

2. Is the manuscript technically sound, and do the data support the conclusions?

Reviewer #3: (No Response)

3. Has the statistical analysis been performed appropriately and rigorously? 

Reviewer #3: (No Response)

4. Have the authors made all data underlying the findings in their manuscript fully available?

Reviewer #3: (No Response)

5. Is the manuscript presented in an intelligible fashion and written in standard English?

Reviewer #3: (No Response)

6. Review Comments to the Author

Reviewer #3: The revised manuscript looks okay to me, and I have no additional comments or suggestions to the authors.

7. PLOS authors have the option to publish the peer review history of their article (what does this mean?). If published, this will include your full peer review and any attached files.

Reviewer #3: No

---

## [Editor Report · Acceptance letter]

6 Nov 2020

PONE-D-20-13875R2 

Factor analysis for the clustering of cardiometabolic risk factors and sedentary behavior, a cross-sectional study. 

Dear Dr. Leu:

I'm pleased to inform you that your manuscript has been deemed suitable for publication in PLOS ONE. Congratulations! Your manuscript is now with our production department. 

Kind regards, 

on behalf of

Dr. Mauro Lombardo 

Academic Editor

PLOS ONE